# Telehealth Exercise Training in Peripheral Arterial Disease (TEXTPAD) study: A pilot randomised controlled trial in socioeconomically disadvantaged populations

James Prentis[1], Arathi Radhakrishnan[1], Eileen Kaner [ID][2], Sandip Nandhra[3], Gerard Stansby[3], Mackenzie Fong[2], Paul Court[4], Gabriel Grizzo Cucato [ID][5]*

1 Department of Perioperative and Critical Care Medicine, Freeman Hospital, Newcastle upon Tyne, United Kingdom, 2 Population Health Sciences Institute, Newcastle University, Newcastle upon Tyne, United Kingdom, 3 Northern Vascular Unit, Freeman Hospital, Newcastle upon Tyne, United Kingdom, 4 Healthworks, Newcastle upon Tyne, United Kingdom, 5 School of Sport, Exercise and Rehabilitation, Northumbria University, Newcastle upon Tyne, United Kingdom

* gabriel.cucato@northumbria.ac.uk

## Abstract

### Background

Peripheral arterial disease (PAD) is a common condition among older adults, particularly those in socioeconomically disadvantaged populations. Supervised exercise is a key treatment for PAD-related claudication, but access to such programs is limited, especially following the COVID-19 pandemic. Telehealth interventions offer a promising alternative, but their feasibility and effectiveness in these populations remain unclear.

### Objective

This study aimed to assess the feasibility of a home-based telehealth intervention combining exercise and behaviour change counselling for patients with PAD from socioeconomically deprived areas of the UK.

### Methods

A pilot randomised controlled trial (RCT) was conducted with PAD patients (n = 36) recruited from the Vascular Unit of Freeman Hospital, Newcastle upon Tyne, UK. Participants were randomly assigned to either a telehealth intervention group (n = 19) or a control group (n = 17). The intervention consisted of weekly phone-based behaviour change counselling and twice-weekly virtual supervised exercise sessions for 12 weeks. Primary feasibility outcomes included recruitment, retention, and adherence. Secondary outcomes assessed functional capacity, quality of life, smoking and alcohol use, and dietary habits.

**Data availability statement:** All relevant data are within the paper and its Supporting Information files.

**Funding:** This work was funded by League of Friends of Newcastle Hospital and Newcastle Hospital's Charity, funding awarded to JP.

## Results

Of the 102 eligible patients, 36 were recruited, falling short of the target recruitment goal of 60 participants. The intervention group attended a median of 20 supervised exercise sessions (max 24) and 11 sessions with the health improvement practitioner (max 12). Exploratory analyses suggested improvements in subjective functional capacity, as measured by walking speed and stair-climbing ability ($P < 0.01$), and quality of life ($P < 0.01$) in the intervention group compared to the control group. A 50% smoking cessation rate was observed among baseline smokers in the intervention group, while no changes were observed in the control group. No significant differences were observed between groups in objective functional capacity (as measured by the 6-minute walk test), alcohol intake, or dietary quality.

## Conclusion

This digitally delivered, home-based telehealth intervention was feasible for PAD patients in socioeconomically disadvantaged areas. Exploratory findings suggest potential benefits in subjective function capacity and health behaviours; however, the study was not powered to evaluate effectiveness, and findings should be interpreted with caution. These data support the suitability of outcome measures and provide essential insights for designing a fully powered definitive trial to evaluate long-term clinical outcomes.

## Introduction

Peripheral arterial disease (PAD) represents a significant global health burden, particularly affecting individuals over the age of 60, with the UK reporting an approximate 20% prevalence [1]. The prevalence is notably higher among socioeconomically disadvantaged populations, who experience more severe forms of the disease, leading to increased disability, amputations, and mortality [2,3]. PAD usually results from the progressive narrowing and hardening of the peripheral arteries due to chronic atherosclerosis, which impede adequate blood flow, especially to the lower extremities. The primary symptom—intermittent claudication—is characterised by discomfort, cramping, or pain during physical activity, typically relieved by rest [4]. This results in reduced functional capacity [5,6], impairment in cardiovascular function [7,8] and diminished quality of life [9]. Notably, those from lower socioeconomic areas often present with more advanced PAD, making early interventions crucial, mainly to reduce cardiovascular risk and prevent disease progression.

The UK's National Institute for Health and Care Excellence (NICE) guidelines recommend supervised exercise as a first-line treatment for claudication, coupled with health behaviour modifications to prevent further cardiovascular events, such as smoking cessation, dietary changes, and weight management [1]. While supervised exercise has been shown to improve walking capacity [10], cardiovascular health [11], and overall quality of life [9], accessibility remains a significant barrier [12].

Health behaviour programs are often costly, underfunded, and particularly difficult to access for disadvantaged patients, who are also affected by additional challenges like reduced walking ability and financial constraints. The COVID-19 pandemic further disrupted the delivery of traditional face-to-face interventions, emphasising the necessity for innovative and accessible approaches.

Telehealth presents a promising, cost-effective alternative to conventional care models, mitigating the physical, financial, and structural barriers to accessing care [13,14]. In this study, telehealth refers to the delivery of healthcare services remotely using information and communication technologies, including video conferencing, telephone calls, and internet-based platforms to support clinical care, health education, and self-management intervention [15]. While the use of telehealth has shown considerable potential the feasibility and efficacy of digital interventions, particularly in resource-limited settings, remain underexplored. Crucially, digital poverty (limited access to internet or devices) and low digital literacy may exclude the very populations most in need of support, potentially widening existing health inequalities. These risks highlight the importance of designing and evaluating digital health interventions in a way that actively addresses issues of inclusion and accessibility.

The telehealth exercise training in PAD (TEXTPAD) study, a pilot feasibility randomised controlled trial, sought to investigate the feasibility and acceptability of a home-based telehealth intervention in improving outcomes in PAD patients residing in deprived areas of Newcastle-upon-Tyne, North Tyneside, Gateshead, and Northumberland. The intervention combined weekly phone-based behavioural counselling with twice-weekly virtual supervised exercise sessions over 12 weeks. Participants were also encouraged to engage in unsupervised walking, supported by activity trackers (Fitbit Charge 4). The study aimed to assess feasibility measures, including recruitment, adherence, and retention, while exploring the preliminary impacts on clinical outcomes, such as functional capacity, lifestyle factors (smoking, alcohol, and diet), and quality of life.

## Materials and methods

### Patient recruitment, randomisation, and sample size

Participants were recruited from the Vascular Unit at Freeman Hospital in Newcastle, UK, from June 11, 2022, to December 31, 2023. Randomisation was performed using an online generator (www.randomizer.org). The target sample size of 60 participants (30 per group) was informed by methodological guidance for pilot trials, which recommends this range to estimate recruitment rates and variance in outcomes needed to inform sample size calculations for a definitive trial [16].

Written informed consent was obtained, and ethical approval was granted by the Research Ethics Committee of Freeman Hospital (April 2021; IRAS Reference No. 286735). The study was registered on ClinicalTrials.gov (NCT05260567).

### Inclusion and exclusion criteria

Patients were eligible if they met the following criteria: a) Ankle-brachial index (ABI) ≤ 0.90; b) Age > 40 years; c) Ability to walk >50 meters, d); Residence in the lowest 30% of super output areas according to the Office for National Statistics (Indices of multiple deprivation score 1–3). Exclusion criteria included the presence of chronic limb-threatening ischemia, very short claudication distance (<50m), severe heart disease (NYHA Grade III or IV), severe stroke, neurodegenerative disease, or uncontrolled hypertension, prior PAD surgery (angioplasty, bypass, etc.) and severe comorbid conditions that would prevent physical activity or study participation

### Interventions

The intervention was developed in collaboration with Newcastle upon Tyne NHS Trust, Northumbria University, Healthworks, and the Newcastle United Foundation. The intervention group received educational materials explaining how

health behaviours affect PAD. Shortly after randomisation, a Health Improvement Practitioner from Healthworks conducted an initial consultation with each participant. The Health Improvement Practitioners (HIPs) from Healthworks are professionally trained in motivational interviewing and behavioural intervention delivery, with qualifications in both nutrition and exercise. Importantly, they are experienced in implementing exercise programs specifically designed for individuals with chronic conditions, including cardiovascular and metabolic diseases. Their extensive background in delivering similar interventions ensured that sessions were conducted safely and effectively. Throughout the program, HIPs provided real-time guidance and adapted exercises as needed based on individual participant tolerance and feedback, helping to support both safety and adherence. Participants who lacked access to a suitable device or internet connection were provided with a tablet and internet access (via SIM card) by the research team to ensure equitable participation in the telehealth intervention.

### Behaviour change program

Participants who were randomised to the intervention group received one-hour, individualised behaviour change sessions via phone or videoconference each week for 12 weeks. Techniques such as goal-setting, problem-solving, and self-regulation were used to modify risk factors. Smokers received cessation support, including nicotine replacement therapy vouchers and e-cigarettes. Alcohol intake was addressed with brief interventions aimed at reducing consumption to low-risk levels (<14 units per week) [17]. Participants also received basic nutrition education based on the British Heart Foundation and Diabetes UK guidelines.

### Supervised exercise

The home-based exercise program was delivered twice weekly via Zoom by the exercise team at Healthworks. Each session included a warm-up (10 minutes), the main exercise (15–20 minutes), and a cool-down (5–10 minutes). The full program details are available elsewhere [18]. The program aimed to improve resistance, aerobic, and functional capacities, such as walking, pushing, pulling, and weight transfer. All sessions were delivered live and supervised by trained professionals, who provided verbal safety instructions and technique feedback. Overall session intensity, including both aerobic and resistance exercises, was monitored using the Borg RPE scale (target: 12–14), which has been used previously in circuit-based training interventions [19]. Participants also received a smartwatch (Fitbit Charge 4, Fitbit Inc, San Francisco, CA) to track step counts and were encouraged to increase their activity by 10% weekly.

### Usual care

Participants in the control group received standard care, including general risk factor modification advice and recommendations from our research team for unsupervised walking (30 minutes, 3–5 times per week). They also received a smartwatch to monitor their physical activity.

### Feasibility outcomes

The feasibility of the intervention was assessed based on patient screening, eligibility, recruitment rates, retention at 12 weeks, and adherence to the intervention (sessions attended and completed).

### Secondary outcomes (Exploratory analysis)

Secondary outcomes related to clinical parameters associated with PAD were collected at Freeman Hospital in Newcastle upon Tyne, with assessments conducted by evaluators blinded to group allocation.

### Objective functional capacity

Objective functional capacity was assessed using the 6-minute walk test (6MWT) at baseline and after 12 weeks. Briefly, participants were encouraged to "walk at their usual pace for six minutes and cover as much ground as possible" and to rest if necessary. The six-minute walking distance was defined as the maximum distance achieved by the patient at the end of the test [20].

### Subjective functional capacity

The Walking Impairment Questionnaire [21] was used to assess subjective walking capacity, including walking distance, speed, and stair-climbing ability. Each domain is anchored from 0, representing extreme limitation, to 100, representing no difficulties. In addition, participants also completed the Walking Estimated Limitation Calculated by History (WELCH) [22] which they were asked how long they could walk at certain speeds, and then how they rated their walking speed relative to their relatives, friends, or people of the same age.

### Quality of life

Generic and specific quality of life were assessed via Euroqol (EQ-5D-5L) and the Vascular Quality of Life Questionnaire-6 (VascuQol-6), respectively. The EQ-5D-5L includes five dimensions: mobility, self-care, usual activities, pain/discomfort, and anxiety/depression. [23]. The VascuQol-6 comprises six items evaluating the impact of vascular disease on social aspects and the capacity to perform daily activities. Each item is scored on a scale of 1–4. The total score is calculated by summarising the scores on each item, resulting in a score between 6 and 24 [24].

### Dietary quality

Dietary quality was assessed using the Short Form Dietary Questionnaire, which collects data on dietary intake frequency and generates a dietary quality score [22].

### Alcohol and tobacco use

The AUDIT-C screening tool [23] assessed alcohol consumption, while carbon monoxide breath tests and self-reported smoking status were used to evaluate tobacco use.

### Statistical analysis

Normality was assessed using the Shapiro-Wilk test, and Levene's test was used to evaluate the homogeneity of variance between groups. Group comparisons for general characteristics were conducted using independent samples t-tests for continuous variables and Pearson's Chi-Square test for categorical variables. For dropout analysis, a Fisher's exact test was used to compare the proportion of participants who withdrew from the intervention and control groups

Descriptive statistics were employed for feasibility outcomes. Secondary outcomes were analysed by comparing delta values (post-intervention mean minus pre-intervention mean) between the intervention and control groups using independent samples t-tests. Statistical significance was set at $P < 0.05$. In addition to reporting absolute mean changes and p-values, standardised mean differences (SMDs; Cohen's *d*) were calculated for all secondary outcomes to provide estimates of intervention effect size.

### Results

Fig 1 illustrates the study flowchart. A total of 102 individuals were assessed for eligibility. Of these, 66 were excluded (39 did not meet the inclusion criteria and 27 declined to participate). Thirty-six participants were randomly assigned, with 19 allocated to the intervention group and 17 to the control group.

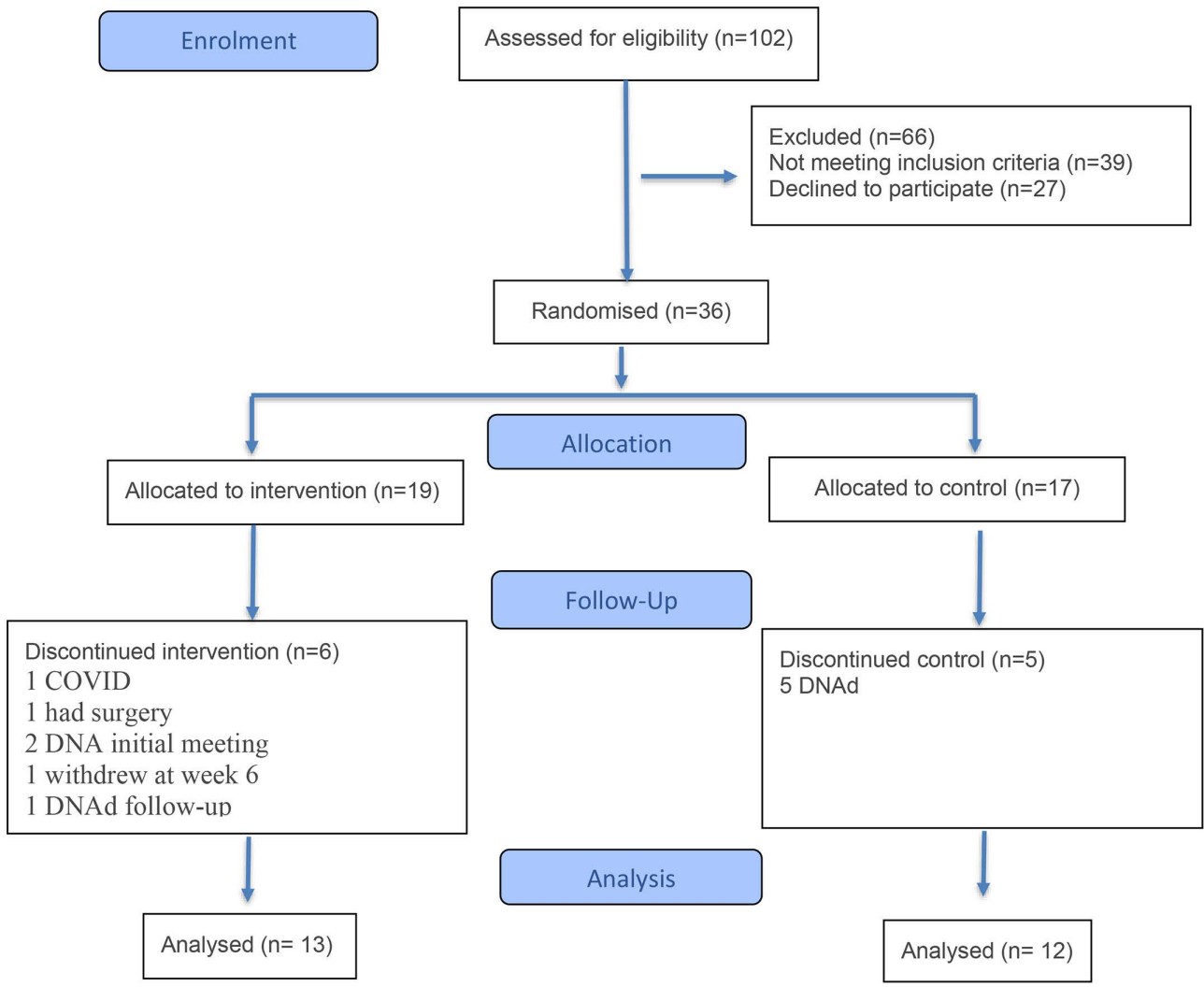

**Fig 1. CONSORT flow diagram showing participant eligibility screening, randomisation, and retention throughout the TEXTPAD pilot trial.**

## Recruitment and retention

Retention was assessed by tracking participant withdrawal and completion rates. In the intervention group, 13 participants (68%) completed the program and follow-up assessments. Reasons for dropout included elective surgery (n = 1), COVID-19 (n = 1), failure to attend the initial assessment (n = 2), and withdrawal at week 6 (n = 1). One additional participant completed the program but missed the final follow-up assessment.

In the control group, 12 participants (71%) completed the trial, while 5 were lost to follow-up due to being uncontactable for post-intervention assessments. A comparison of dropout rates revealed no significant difference between the intervention (6/19; 31.6%) and control (5/17; 29.4%) groups (Fisher's exact test, P = 1.00), suggesting that attrition was not group-dependent. The final sample included 25 participants (13 in the intervention group and 12 in the control group) for feasibility and exploratory analysis.

## Adherence to the telehealth intervention

Adherence to the telehealth program was assessed by evaluating participation in supervised exercise sessions and health improvement practitioner sessions over the 12-week period. A total of 24 supervised exercise sessions were expected. Among participants who completed the intervention, the median number of sessions attended was 20 (range: 0–24). Three participants achieved adherence above 90%, four between 75–90%, and three between 50–75%. Two participants did not attend any exercise sessions—one declined due to housing issues, and the other cited personal reasons. For the health behaviour intervention, a total of 12 sessions (once per week) were expected. The median number of sessions attended was 11 (range: 7–12). Ten participants (77%) had adherence above 90%, two (15%) attended between 75–90% of sessions, and one (8%) attended between 50–75% of sessions.

## Accessibility to the telehealth intervention

Accessibility was evaluated based on internet access and device availability. Both groups had similar internet access, with two participants in each group reporting no internet access. Regarding device availability, two participants in the intervention group and one in the control group did not own a suitable device. Connectivity issues were minimal, with only one participant missing an exercise session due to technical difficulties.

Table 1 shows the clinical characteristics, smoking status, alcohol, diet score, quality of life, and objective and subjective measurements of functional capacity. The mean age was 65.0 (2.4) years in the control group and 66.8 (1.0) years in the intervention group ($P = 0.49$). In terms of gender distribution, the control group consisted of 9 males (75%) and 3 females (25%), while the intervention group comprised 11 males (85%) and 2 females (15%).

We did not observe any differences between the two groups regarding clinical characteristics. In addition, no differences were observed at baseline for smoking status, alcohol intake, diet score, specific and general quality of life, and objective and subjective measurements of functional capacity.

Table 2 shows the comparison of the delta values (post-pre values) for the secondary outcomes between the intervention and control groups. We did not observe significant statistical differences between groups for 6MWT, general quality of life (EQ5D), and nutritional score. However, we observed a significant increase in the intervention group compared to the control in Walking Impairment Questionnaire WIQ speed ($\Delta$ intervention $= 22.2 \pm 4.6$ vs $\Delta$ control $= 5.5 \pm 4.9$, $P = 0.002$), WIQ stair ($\Delta$ intervention $= 20.0 \pm 7.0$ vs $\Delta$ control $= -5.6 \pm 5.3$, $P = 0.001$) and specific disease quality of life VascuQoL-6 ($\Delta$ intervention $= 3.7 \pm 0.81$ vs $\Delta$ control $= 0.41 \pm 0.95$, $P = 0.01$).

Six participants were smoking in the intervention group and 8 in the control group at baseline (a total of 56% current smokers). We observed that 3 participants in the intervention group stopped smoking (50% quit rate) at follow-up, whilst all participants continued to smoke in the control group.

Risky alcohol use (audit C score >8) was high in both the control (25%) and intervention groups (33%) at baseline. There were no significant changes in alcohol consumption for either group at follow-up. In the control group, 8 of 13 (67%) participants had the same alcohol consumption score; 2 (16.7%) increased, whereas the other 2 (16.7%) reduced. In the intervention group, 9 of 13 (69.3%) participants had the same; 2 (23%) increased, and only 1 patient (7.6%) reduced their alcohol status.

## Discussion

This study aimed to assess the feasibility of implementing a home-based telehealth intervention focused on health behaviour modification for patients with PAD living in socioeconomically deprived areas of the North East of England, while also exploring the preliminary impact of the intervention on clinical outcomes. Recruitment and retention challenges were anticipated, given the target population's socioeconomic vulnerabilities and the additional barriers posed by the COVID-19 pandemic. Despite these difficulties, participant engagement and retention rates indicated that the telehealth-based intervention was feasible in those who were recruited. Of the 13 participants randomised to the

**Table 1. Baseline general data of control and intervention groups.**

|  | Control (N = 19) | Intervention (N = 17) | P |
|---|---|---|---|
| Age | 65.0 (2.4) | 66.8 (1.0) | 0.49 |
| Sex (%) | Male (85) Female (15) | Male (75) Female (25) | 0.92 |
| BMI (kg/m$^2$) | 29.1 (5.72) | 27.3 (6.90) | 0.42 |
| BMI > 30 kg/m$^2$ (%) | 33 | 46 | 0.89 |
| *Medication* |  |  |  |
| Antiplatelet (%) | 100 | 100 | 1.00 |
| Statins (%) | 100 | 100 | 1.00 |
| Inhibitor of ACE (%) | 23.5 | 42.1 | 0.23 |
| Beta blockers (%) | 35.3 | 47.4 | 0.46 |
| **Connectivity** |  |  |  |
| No internet (%) | 17 | 15 | 0.92 |
| No device (%) | 8 | 15 | 0.58 |
| *Smoking* Current (%) Ex (%) None (%) | 66.7 25 8.3 | 46.1 53.8 0 | 0.24 |
| *Alcohol* Audit C 0–3 (%) Audit C 4–8 (%) Audit C above 8 (%) | 41.7 33.3 25.0 | 30.8 38.5 30.8 | 0.85 |
| Dietary score *Quality of Life* | 9.75 (1.5) | 10.5 (1.6) | 0.21 |
| EQ5D utility score | 0.410 (.252) | 0.419 (0.252) | 0.93 |
| EQ5D VAS score | 42.2 (19) | 44.7 (24) | 0.78 |
| VascuQoL-6 | 11.3 (1.0) | 11.2 (0.9) | 0.94 |
| **Function Capacity** |  |  |  |
| 6MWT (m) | 289.2 (56) | 249 (139) | 0.37 |
| WELCH | 17.9 (2.9) | 14.3 (4.2) | 0.51 |
| WIQ distance | 44.3 (5.4) | 47.9 (6.3) | 0.67 |
| WIQ speed | 34.4 (6.5) | 27.4 (5.6) | 0.42 |
| WIQ stairs | 37.4 (9.0) | 52.0 (9.9) | 0.29 |

Data is present in mean ± standard error or frequency. BMI – body mass index; 6MWT – six-minute walking test; WELCH – Walking Estimated Limitation Calculated by History Questionnaire; WIQ – Walking Impairment Questionnaire; ACE – Angiotensin-converting enzyme.

intervention group, a median of 20 out of a possible 24 supervised exercise sessions were completed, and a median of 11 out of 12 lifestyle change sessions were attended over 12 weeks.

However, it is important to consider the characteristics of those who dropped out. In the intervention group, five participants did not complete the study. Reasons included elective surgery (n = 1), COVID-19 infection (n = 1), failure to attend the initial assessment (n = 2), and voluntary withdrawal at week 6 (n = 1). These dropouts may reflect the complex and often unstable life circumstances of participants in disadvantaged communities, including competing health priorities and difficulties with consistent engagement. In addition, a formal comparison using Fisher's exact test revealed no significant difference in dropout rates between groups (intervention: 6/19; control: 5/17; P = 1.00). Although these data were limited, they highlight the need for future studies to incorporate more detailed analyses of attrition and explore strategies to improve engagement and reduce barriers to continued participation.

**Table 2. Delta values (post-pre) for control and intervention groups.**

| | Δ Control (n = 12) | Δ Intervention (n = 13) | P | SMD (Cohen's D) |
|---|---|---|---|---|
| **QUALITY OF LIFE** | | | | |
| EQ5D utility score | 0.07 (0.19) | 0.16 (0.15) | 0.28 | 0.16 |
| EQ5D VAS score | 12.1 (0.9) | 10.6 (0.9) | 0.27 | 0.49 |
| VascuQoL-6 | 0.41 (0.95) | 3.7 (0.81) | **0.01** | **1.11** |
| **FUNCTIONAL CAPACITY** | | | | |
| 6MWT (m) | 67.0 (16.6) | 95.0 (26.7) | 0.39 | 0.38 |
| WELCH total score | −0.16 (2.7) | 11.1 (4.9) | **0.05** | **0.82** |
| WIQ distance | 8.2 (2.9) | 15.9 (5.6) | 0.25 | 0.50 |
| WIQ speed | 5.5 (4.9) | 22.2 (4.6) | **0.02** | **1.04** |
| WIQ stairs | −5.5 (5.3) | 20.0 (7.0) | **0.01** | **1.22** |
| **DIETARY SCORE** | | | | |
| Dietary score | 0.0 (1.35) | 0.77 (1.4) | 0.17 | 0.16 |

Data is present in mean ± standard error. 6MWT – six-minute walking test; WELCH – Walking Estimated Limitation Calculated by History Questionnaire; WIQ – Walking Impairment Questionnaire; Standardised Mean Difference.

Overall, these findings support the feasibility of a remote intervention in this population, underscoring its potential as an accessible and cost-effective alternative to traditional in-person supervised exercise programs.

The health needs of the PAD population in socioeconomically disadvantaged areas are particularly pronounced. A significant proportion of the sample were smokers (56%), 40% had obesity (BMI > 30 kg/m$^2$), and many were identified as risky drinkers, Audit C score >8 (28%). These figures reflect the acute health risks faced by this population, reinforcing the need for targeted interventions to address lifestyle factors such as sedentary behaviour, smoking, obesity, and alcohol consumption [25]. Furthermore, the baseline quality of life scores for these participants, as measured by the EQ-5D-5L, were substantially lower than those reported for a general PAD population [26], highlighting the particularly poor health status of those from our sample living in socioeconomically deprived areas. These findings emphasise the urgent need for tailored interventions to address the specific challenges faced by PAD patients in disadvantaged communities.

The secondary objective of this pilot trial was to conduct exploratory analyses evaluating the potential clinical effectiveness of the intervention to inform the design of a future definitive trial. Statistically significant improvements were observed in subjective functional outcomes, specifically in self-reported walking speed and stair-climbing ability, as measured by the Walking Impairment Questionnaire (WIQ). These results may indicate a beneficial impact of the telehealth-based exercise and behaviour change intervention on perceived functional limitations commonly associated with PAD. Additionally, the intervention group demonstrated significant improvements in vascular-specific quality of life, as assessed by the VascuQoL-6 instrument, relative to the control group.

However, interpretation of these findings must be undertaken with caution. The trial was not powered to evaluate clinical efficacy, and a single prespecified primary outcome was not defined, thereby limiting the ability to distinguish true intervention effects from random variation. Furthermore, improvements were not observed in objective measures of functional capacity (i.e., the 6 MWT), which may reflect either limited statistical power or the influence of physical activity advice and device provision in the control group. These findings underscore the importance of selecting sensitive and clinically meaningful outcome measures in future trials. In addition, although no adverse events were formally reported, the study did not include a systematic evaluation of safety outcomes. Given that PAD patients often present with multiple comorbidities and elevated cardiovascular risk, future studies must incorporate structured adverse event monitoring to ensure intervention safety and acceptability. Attrition was also notable, with several participants withdrawing before completing follow-up assessments. While some reasons for discontinuation were documented (e.g., elective surgery, COVID-19 infection),

understanding the characteristics and motivations of those who withdrew will be critical in optimising retention strategies in larger-scale evaluations. It is also important to note that no formal baseline exercise testing (e.g., treadmill or cardiopulmonary exercise testing) was conducted to prescribe individualised exercise intensity. Instead, the program followed a pragmatic circuit training model that combined aerobic and resistance-based multijoint exercises at low to moderate intensity, guided by the Borg RPE scale (target range: 12–14). This design was chosen to prioritise safety, feasibility, and accessibility in a remote format. While no exercise-related side effects or adverse events were observed, we acknowledge that the absence of individualised exercise prescription is a limitation. Incorporating baseline exercise assessments in future trials may help tailor intensity more precisely and further enhance both safety monitoring and intervention effectiveness.

In summary, these exploratory findings provide initial evidence supporting the potential of digitally delivered interventions for PAD management. Nonetheless, substantial uncertainties remain regarding clinical effectiveness, long-term outcomes, and implementation at scale. A fully powered, methodologically rigorous trial is warranted to determine the efficacy, cost-effectiveness, and safety of such interventions, including the identification of a core outcome set and strategies to mitigate digital exclusion, maximise engagement, and monitor adverse events. These elements are essential for informing commissioning decisions and supporting future service delivery in socioeconomically disadvantaged populations. In addition, in this pilot feasibility study, missing data were handled using a complete case approach, and no imputation or formal intention-to-treat analysis was conducted. While this is consistent with the exploratory nature of feasibility trials, it limits the generalisability and robustness of outcome findings. A future definitive trial should incorporate predefined strategies for handling missing data and non-adherence, including intention-to-treat analysis and appropriate methods such as multiple imputation, to improve methodological rigour.

One of the most notable findings was the significant reduction in smoking prevalence in the intervention group, with 50% of baseline smokers reporting cessation at follow-up, compared to no change in the control group. This reinforces the value of incorporating smoking cessation support into interventions for PAD management, which has the potential to improve long-term outcomes in this high-risk population [27]. In contrast, changes in alcohol consumption were minimal. Although one participant in the intervention group reduced their intake, two participants in the control group also reported reductions, and others in both groups showed no change or increased consumption. This highlights a potential gap in the intervention's effectiveness and suggests that further tailored strategies may be necessary to address alcohol use in PAD patients more effectively. It may also be useful to explore the integration of more intensive alcohol-related interventions, such as counselling, within future iterations of the program [28].

The results of this study align with the NICE guidelines [1], which advocate supervised exercise as the first-line treatment for PAD. This study demonstrates that a home-based telehealth delivery model could complement traditional approaches and provide an alternative for patients with PAD in underserved areas. Importantly, this new model incorporates health behaviour changes, which are also recommended by NICE guidelines, and addresses common barriers to in-person care, such as transportation and mobility issues, that disproportionately affect socioeconomically disadvantaged populations. These findings suggest that home-based telehealth interventions could be crucial in enhancing access to PAD management, particularly for individuals who face significant challenges in accessing traditional healthcare services.

While this study provides valuable insights, we recognise that there are several limitations. The small sample size and difficulties in recruitment limit the study's statistical power and restrict the generalisability of its findings. More work may be needed to focus the intervention on a larger proportion of eligible patients and to include a definitive experimental design study to evaluate effectiveness and cost-effectiveness. Attrition occurred in both study arms. While reasons for withdrawal were documented in several cases, the potential role of digital exclusion—defined as limited access to, or ability to use, internet-enabled devices—must be acknowledged. Given the potential for digital interventions to inadvertently widen health inequalities, future trials must incorporate robust digital access, literacy, and confidence assessments to identify and address barriers to participation and sustained engagement. Additionally, the lack of long-term follow-up means we cannot assess the sustainability of the observed improvements or their impact on clinical outcomes such as functional

capacity, cardiovascular events and mortality. Future studies with larger sample sizes and extended follow-up periods are needed to confirm the long-term effectiveness and clinical impact of telehealth interventions in patients with PAD. Furthermore, the reliance on self-reported data for alcohol consumption and smoking habits introduces the potential for reporting bias, and future studies should consider incorporating objective measures, such as biomarkers or digital tracking, to reduce this risk. Technical issues during the study prevented the collection of physical activity data, limiting our ability to fully evaluate the intervention's impact on lifestyle behaviours. Future research should prioritise resolving technological challenges to ensure comprehensive data collection, including physical activity metrics.

In terms of improving the intervention, future studies could also explore the inclusion of more personalised components, such as tailored dietary counselling or more intensive alcohol interventions, to address other modifiable risk factors in PAD patients.

## Conclusion

In conclusion, the TEXTPAD study provides preliminary evidence supporting the feasibility and potential positive effects of a home-based telehealth intervention in PAD patients from socioeconomically deprived areas. The significant improvements in subjective functional capacity, quality of life, and smoking cessation observed in the intervention group highlight the potential benefits of such programs. These findings contribute to the growing body of evidence supporting the use of home-based telehealth interventions for PAD management, particularly in underserved populations. Moving forward, further research is required to optimise the intervention, enhance adherence, and evaluate the long-term clinical impact of telehealth interventions in PAD patients, focusing on improving outcomes such as cardiovascular events and mortality.

## Supporting information

**S1 File.  CONSORT 2025 Checklist TEXTPAD.**
(DOCX)

**S2 FIle.  Copy of textpad results.xlsx so com quem termniou.**
(XLSX)

**S3 File.  TEXT-Pad+protocol v1.6 clean (1).**
(PDF)

## Acknowledgments

We would like to acknowledge the League of Friends of Newcastle Hospital and Newcastle Hospital's Charity for funding the study, the vascular research team for their help and support, and the staff at Healthworks for engaging the participants in the study.

## Author contributions

**Conceptualization:** James Prentis, Arathi Radhakrishnan, Eileen Kaner, Sandip Nandhra, Gerard Stansby, Mackenzie Fong, Paul Court, Gabriel Grizzo Cucato.

**Data curation:** Mackenzie Fong.

**Formal analysis:** James Prentis, Gabriel Grizzo Cucato.

**Funding acquisition:** James Prentis.

**Methodology:** Eileen Kaner, Gerard Stansby, Mackenzie Fong, Gabriel Grizzo Cucato.

**Project administration:** Gabriel Grizzo Cucato.

**Writing – original draft:** James Prentis, Arathi Radhakrishnan, Eileen Kaner, Sandip Nandhra, Gerard Stansby, Mackenzie Fong, Paul Court, Gabriel Grizzo Cucato.

**Writing – review & editing:** James Prentis, Arathi Radhakrishnan, Eileen Kaner, Sandip Nandhra, Gerard Stansby, Mackenzie Fong, Paul Court, Gabriel Grizzo Cucato.

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
