## [Decision Letter · Decision Letter 0]

28 Jul 2025

Dear Dr. Cucato,

Thank you for submitting your manuscript to PLOS ONE. After careful consideration, we feel that it has merit but does not fully meet PLOS ONE’s publication criteria as it currently stands. Therefore, we invite you to submit a revised version of the manuscript that addresses the points raised during the review process.

Please define telehealth in the introduction since other readers may not know what telehealth is.

We look forward to receiving your revised manuscript.

Kind regards,

Enock Madalitso Chisati, PhD

Academic Editor

PLOS ONE

Journal Requirements:

2. We note that you have selected “Clinical Trial” as your article type. PLOS ONE requires that all clinical trials are registered in an appropriate registry (the WHO list of approved registries is at https://www.who.int/clinical-trials-registry-platform/network/primary-registries" https://www.who.int/clinical-trials-registry-platform/network/primary-registries and more information on trial registration is at http://www.icmje.org/about-icmje/faqs/clinical-trials-registration/). Please state the name of the registry and the registration number (e.g. ISRCTN or ClinicalTrials.gov) in the submission data and on the title page of your manuscript. a) Please provide the complete date range for participant recruitment and follow-up in the methods section of your manuscript. b) If you have not yet registered your trial in an appropriate registry, we now require you to do so and will need confirmation of the trial registry number before we can pass your paper to the next stage of review. Please include in the Methods section of your paper your reasons for not registering this study before enrolment of participants started. Please confirm that all related trials are registered by stating: “The authors confirm that all ongoing and related trials for this drug/intervention are registered”. Please see http://journals.plos.org/plosone/s/submission-guidelines#loc-clinical-trials for our policies on clinical trials.

3. Thank you for stating the following in the Acknowledgments Section of your manuscript: [We would like to acknowledge the League of Friends of Newcastle Hospital and Newcastle Hospital’s Charity for funding the study, the vascular research team for their help and support, and the staff at Healthworks for engaging the patients in the study.

Please remove any funding-related text from the manuscript and let us know how you would like to update your Funding Statement. Currently, your Funding Statement reads as follows: “The authors received no specific funding for this work.”

4. Please include captions for your Supporting Information files at the end of your manuscript, and update any in-text citations to match accordingly. Please see our Supporting Information guidelines for more information: http://journals.plos.org/plosone/s/supporting-information .

Additional Editor Comments:

Define telehealth in the introduction of the manuscript.

Reviewers' comments:

Reviewer's Responses to Questions

**Comments to the Author**

1. Is the manuscript technically sound, and do the data support the conclusions?

Reviewer #1: Yes

Reviewer #2: Partly

2. Has the statistical analysis been performed appropriately and rigorously?

Reviewer #1: Yes

Reviewer #2: Yes

3. Have the authors made all data underlying the findings in their manuscript fully available?

Reviewer #1: Yes

Reviewer #2: No

4. Is the manuscript presented in an intelligible fashion and written in standard English?

Reviewer #1: Yes

Reviewer #2: Yes

Reviewer #1: The research team recruited 37 PAD patients for a RCT to assess the feasibility and acceptability of a home-based telehealth intervention. They concluded its feasibility and acceptability for PAD patients in socioeconomically disadvantaged areas with potential benefits in subjective function capacity and health behaviors.

1. Please provide a brief rationale or reference for the target sample size of 60 or effect size estimation.

2. While the dropout rates are comparable between groups, it would be informative to report an official comparison on whether they were likely to randomly withdraw.

3. How to handle the missing data in the analysis, e.g., those without achieving full adherence?

4. The actual recruited sample size is much smaller than the expected sample size, which may affect the interpretation of the results. With this sample size, standardized mean differences or relevant information may be provided as an information for future trials planning.

Reviewer #2: SUMMARY

This study aimed to assess feasibility and acceptability of a Tele-Health intervention, which constituted lifestyle modification targeted educative reading materials, videos and one-one interaction with specially trained health improvement practitioners. Participants attended a phone-based educative/counseling session once a week via a phone call and participated in a zoom-based exercise therapy twice a week. The control intervention constituted the usually offered lifestyle education and counseling aimed at increasing physical activity levels (e.g doing 30 minutes of exercise three times a week) and adoption of healthy behaviors, which are offered at the PAD clinic. A total of n=37 participants living with PAD were recruited, instead of an expected total of n=60 participants. An exploratory analytical approach was conducted to answer the study’s set objectives.

ABSTRACT

1. Authors conclude that the intervention is ‘feasible and acceptable’. In the study protocol, it is clearly indicated that qualitative interviews were (possibly) done to explore the experiences of participants after receiving the intervention, to determine the ‘acceptability’ of the program. Nevertheless, this aspect has not been reported in the study methods of this paper (measures, analytical plan) and results sections; yet it appears in the abstract’s conclusion. How did the authors draw this conclusion?

INTRODUCTION

1. Page 3: line 10. Authors should consider putting a space between the ‘quality of life’ and (9)

2. Page 3: Line 14. ‘and weight management (1). Authors should consider putting a space between ‘management and (1).

3. Pages 3-4: Authors describe the aim of the study, which are mainly focusing on assessing the feasibility and ‘acceptability’ of the Telehealth intervention. However, the subsequent sections (including results section) have not narrated the assessment of acceptability aspect, and why it was important to assess. It may be important to insert such information to indicate the relevance of the approach/intervention.

METHODS

Inclusion and exclusion criteria

1. Authors indicate that they enrolled participants aged <40 years. Did this age limit contribute to the low turnover rate?

a. Why limiting the age to <40 years when in the introduction it is mentioned that PAD particularly affect those aged over 60 years?

2. Were the patients on secondary prevention medications? This has not been mentioned in the protocol either.

3. Authors have also reported that patients with ‘chronic limb-threatening ischemia’ were excluded. Was this based on a known existing diagnosis?

Intervention

1. Authors report that ‘prescribed exercise’ was done at home (intervention arm). It is not clear if the participants underwent baseline screening to establish their appropriate exercise target zone (for both safety and effectiveness of the intervention)?

2. Authors also mention that the intensity was prescribed and progressed between 12-14 Borg-RPE-20 scale. The intervention comprised of both aerobic and strength exercises, were the strength exercises prescribed? Considering that Bog-RPE scale is suitable for monitoring aerobic capacity. How did authors consider safety of the participants in using ‘household objects’ for strength training if no personalised prescription was done?

3. I am assuming that all participants had access to mobile phones (smart phones) to access all intervention materials? Or were they supported with such resources (the 8% and 15% with no device/internet access)?

4. Are the health improvement practitioners also trained in ‘exercise prescription’ or they are promoted to implement already prescribed exercise therapy? Is the dream to have a partially structured physical activity intervention in this case?

RESULTS AND DISCUSSION

1. On page 7, results section, authors indicate that there were n=99 eligible participants, yet the flowchart indicate n=60 as passing the pre-set inclusion criteria.

2. Due to the limited interpretive nature of the quantitative descriptive data, authors may consider presenting the qualitative data drawn from the qualitative individual interviews aimed at assessing the acceptability of the intervention. Hearing from the participants may be a better supplementary information, key to understanding the technicalities of the intervention.

GENERAL COMMENTS

1. Authors refer to study participants as ‘patients’. To reflect patients’ active role and contribution towards the research process (compared to their passiveness in clinical care ‘delivery-access’ processes) it is generally recommended to refer to them as ‘participants’ in research. Authors may wish to reconsider this.

**Do you want your identity to be public for this peer review?** For information about this choice, including consent withdrawal, please see our Privacy Policy

Reviewer #1: No

Reviewer #2: No

---

## [Author Response · Author response to Decision Letter 1]

11 Sep 2025

We would like to sincerely thank the reviewers and the academic editor for their thoughtful and constructive feedback on our manuscript. We greatly appreciate the time and effort invested in evaluating our work. The comments have been invaluable in helping us clarify and strengthen the manuscript. Below, we provide detailed point-by-point responses to each comment, indicating the changes made in the revised version. We hope the revised manuscript now meets the standards for publication in PLOS ONE.

Journal Requirements:

1. Please ensure that your manuscript meets PLOS ONE's style requirements, including those for file naming. The PLOS ONE style templates can be found at https://journals.plos.org/plosone/s/file?id=wjVg/PLOSOne_formatting_sample_main_body.pdf nd https://journals.plos.org/plosone/s/file?id=ba62/PLOSOne_formatting_sample_title_authors_affiliations.pdf

ANSWER: Done

2. We note that you have selected “Clinical Trial” as your article type. PLOS ONE requires that all clinical trials are registered in an appropriate registry (the WHO list of approved registries is at https://www.who.int/clinical-trials-registry-platform/network/primary-registries" https://www.who.int/clinical-trials-registry-platform/network/primary-registries and more information on trial registration is at http://www.icmje.org/about-icmje/faqs/clinical-trials-registration/). Please state the name of the registry and the registration number (e.g. ISRCTN or ClinicalTrials.gov) in the submission data and on the title page of your manuscript. a) Please provide the complete date range for participant recruitment and follow-up in the methods section of your manuscript. b) If you have not yet registered your trial in an appropriate registry, we now require you to do so and will need confirmation of the trial registry number before we can pass your paper to the next stage of review. Please include in the Methods section of your paper your reasons for not registering this study before enrolment of participants started. Please confirm that all related trials are registered by stating: “The authors confirm that all ongoing and related trials for this drug/intervention are registered”. Please see http://journals.plos.org/plosone/s/submission-guidelines#loc-clinical-trials for our policies on clinical trials.

ANSWER: We confirm that the study was prospectively registered on ClinicalTrials.gov before participant enrolment. The registration number has now been added to the title page and submission metadata as requested. We also confirm that all ongoing and related trials for this intervention are registered.

Please remove any funding-related text from the manuscript and let us know how you would like to update your Funding Statement. Currently, your Funding Statement reads as follows: “The authors received no specific funding for this work.”

ANSWER: Thanks. We removed this information from the text and updated the funding statement.

ANSWER: Thanks. We updated in the new version of the manuscript.

Additional Editor Comments:

Define telehealth in the introduction of the manuscript.

ANSWER: We included it in the new version of the manuscript.

Reviewer 1

1. Please provide a brief rationale or reference for the target sample size of 60 or effect size estimation.

ANSWER: Thank you for this helpful comment. The target sample size of 60 participants (30 per group) was based on published recommendations for pilot and feasibility trials, which aim to estimate parameters necessary for designing a future definitive trial. Specifically, Lancaster et al. (2004) suggest that a sample size between 24–50 per group is adequate to estimate key feasibility outcomes such as recruitment, retention, and variance in outcome measures. We chose 60 as a pragmatic and slightly conservative target, considering expected attrition and the socioeconomically disadvantaged nature of the target population. This information can be found in the "Patient Recruitment, Randomisation, and Sample Size" section of the manuscript and is cited in reference 15.

2. While the dropout rates are comparable between groups, it would be informative to report an official comparison on whether they were likely to randomly withdraw.

ANSWER: Thank you for this important point. We have now conducted a formal comparison of dropout rates between the intervention (6/19; 31.6%) and control (5/17; 29.4%) groups using Fisher’s exact test, which is appropriate for small samples. The result was non-significant (P = 1.00), indicating no statistically significant difference in attrition between groups. This suggests that withdrawals were likely unrelated to group allocation. We have added this information to the Results – Recruitment and Retention section and addressed its implication in the Discussion section. Added to Results: “A comparison of dropout rates showed no significant difference between the intervention (6/19; 31.6%) and control (5/17; 29.4%) groups (Fisher’s exact test, P = 1.00). (page 8). Added to Discussion: “Formal analysis of dropout rates between groups revealed no significant difference, indicating that withdrawal was likely random and not attributable to intervention-specific factors. (page 9).

3. How to handle the missing data in the analysis, e.g., those without achieving full adherence?

ANSWER: As this was a pilot feasibility trial, the primary aim was to assess the practicality of recruitment, retention, and intervention delivery, rather than to evaluate clinical effectiveness. Therefore, outcome analyses were exploratory and based on complete case analysis, including only participants who completed both baseline and follow-up assessments.

Regarding adherence, we note that two participants in the intervention group did not attend any exercise sessions due to personal issues (e.g., housing instability and caregiving responsibilities). However, both individuals fully participated in the behaviour change counselling sessions, which are a core component of the intervention. Given the small sample size and the pilot nature of the study, we chose to retain these participants in the exploratory analysis to reflect real-world variability in engagement and to avoid further reducing statistical power.

In addition, a formal intention-to-treat analysis and imputation for missing data were not conducted, as the study was not powered to detect efficacy. However, we acknowledge this as a limitation. We included this information in the discussion section (Page 12) “In addition, in this pilot feasibility study, missing data were handled using a complete case approach, and no imputation or formal intention-to-treat analysis was conducted. While this is consistent with the exploratory nature of feasibility trials, it limits the generalisability and robustness of outcome findings. A future definitive trial should incorporate predefined strategies for handling missing data and non-adherence, including intention-to-treat analysis and appropriate methods such as multiple imputation, to improve methodological rigour..”

4. The actual recruited sample size is much smaller than the expected sample size, which may affect the interpretation of the results. With this sample size, standardized mean differences or relevant information may be provided as an information for future trials planning.

ANSWER. Based on your comment, we calculated the standardised mean differences (SMDs/Cohen’s D) for the secondary outcomes in Table 2, alongside the delta values and p-values. This was updated in Table 2 of the new manuscript.

Reviewer 2

ABSTRACT

1. Authors conclude that the intervention is ‘feasible and acceptable’. In the study protocol, it is clearly indicated that qualitative interviews were (possibly) done to explore the experiences of participants after receiving the intervention, to determine the ‘acceptability’ of the program. Nevertheless, this aspect has not been reported in the study methods of this paper (measures, analytical plan) and results sections; yet it appears in the abstract’s conclusion. How did the authors draw this conclusion?

ANSWER: We would like to thank you for your question. We deleted this information since our qualitative analysis was conducted in a separate study, which has recently been published (please see https://www.sciencedirect.com/science/article/pii/S2949912724001235).

INTRODUCTION

1. Page 3: line 10. Authors should consider putting a space between the ‘quality of life’ and (9)

ANSWER: Done

2. Page 3: Line 14. ‘and weight management (1). Authors should consider putting a space between ‘management and (1).

ANSWER: Done

3. Pages 3-4: Authors describe the aim of the study, which are mainly focusing on assessing the feasibility and ‘acceptability’ of the telehealth intervention. However, the subsequent sections (including results section) have not narrated the assessment of acceptability aspect, and why it was important to assess. It may be important to insert such information to indicate the relevance of the approach/intervention.

ANSWER: We would like to thank you for your question. We deleted this information since our qualitative analysis was conducted in a separate study, which has recently been published (please see here https://www.sciencedirect.com/science/article/pii/S2949912724001235).

METHODS

Inclusion and exclusion criteria

1. Authors indicate that they enrolled participants aged <40 years. Did this age limit contribute to the low turnover rate?

a. Why limiting the age to <40 years when in the introduction it is mentioned that PAD particularly affect those aged over 60 years?

ANSWER: Thank you for both questions. It was our mistake. The inclusion criteria were age over 40 years, since PAD is more prevalent in elderly individuals. We have updated this information in the new version of the manuscript.

2. Were the patients on secondary prevention medications? This has not been mentioned in the protocol either.

ANSWER: Thank you for your question. We have combined the medication data from both groups and included a formal analysis of this information. This update is reflected in the new version of the manuscript.

3. Authors have also reported that patients with ‘chronic limb-threatening ischemia’ were excluded. Was this based on a known existing diagnosis?

ANSWER: Thank you for your question. Yes, the exclusion of patients with chronic limb-threatening ischemia was based on both clinical records and assessments performed during recruitment. The research team verified the absence of chronic limb-threatening ischemia through review of the patient’s medical records, clinical vascular examination, and ankle-brachial index (ABI) measurement. Patients with clinical signs of rest pain, tissue loss, or ABI < 0.40, which is indicative of critical limb ischemia, were not included in the study.

Intervention

1. Authors report that ‘prescribed exercise’ was done at home (intervention arm). It is not clear if the participants underwent baseline screening to establish their appropriate exercise target zone (for both safety and effectiveness of the intervention)?

ANSWER: Thank you for this important point. We did not perform a baseline exercise testing (e.g., treadmill or CPET) to prescribe individualised exercise intensity and or measure the effectiveness of the intervention. However, the exercise program was deliberately designed as a low-to-moderate-intensity circuit training, combining aerobic and resistance components using multijoint functional movements. This structure aimed to ensure accessibility, safety, and feasibility for patients with peripheral arterial disease. Intensity was monitored using the Borg RPE scale (target: 12–14, “somewhat hard”). Importantly, no adverse events or exercise-related side effects were observed in any participant. We acknowledge that the lack of objective maximal testing is a limitation that may have affected the precision of exercise prescription, and we have added this to the Discussion section of the manuscript.

2. Authors also mention that the intensity was prescribed and progressed between 12-14 Borg-RPE-20 scale. The intervention comprised of both aerobic and strength exercises, were the strength exercises prescribed? Considering that Bog-RPE scale is suitable for monitoring aerobic capacity. How did authors consider safety of the participants in using ‘household objects’ for strength training if no personalised prescription was done?

ANSWER: Thank you for this valuable comment. Yes, strength exercises were prescribed as part of the structured circuit training sessions. The sessions were designed by qualified exercise professionals and included both aerobic and multijoin resistance exercises. These were delivered in real time via Zoom, with adaptations based on participant feedback and functional ability.

The Borg RPE scale (12–14) was used to monitor overall perceived exertion during the entire circuit session, including both aerobic and resistance components. While it is traditionally used in aerobic training, previous studies have applied the Borg RPE scale to monitor intensity during combined or circuit-based training, which often includes both modalities. We have now clarified this in the Methods section and added a note to the Discussion acknowledging the limitation regarding individual strength prescription.

For resistance exercises, we used functional movements (e.g., sit-to-stand, push/pull exercises) with bodyweight or light household items (e.g., water bottles, canned goods). All exercises were demonstrated and supervised in real time, and verbal safety guidance was provided. No adverse events or safety concerns were reported. However, we acknowledge that the absence of objective baseline strength assessment and personalised load progression may be a limitation in optimising the resistance component. Added to Methods, Supervised Exercise section:

“All sessions were delivered live and supervised by trained professionals, who provided verbal safety instructions and technique feedback. Overall session intensity, including both aerobic and resistance exercises, was monitored using the Borg RPE scale (target: 12–14), which has been used previously in circuit-based training interventions.”

3. I am assuming that all participants had access to mobile phones (smart phones) to access all intervention materials? Or were they supported with such resources (the 8% and 15% with no device/internet access)?

ANSWER: Thank you for this important comment. While most participants had access to their own smartphone, tablet, or computer with an internet connection, participants who lacked access to suitable devices or internet were fully supported by the research team. Specifically, tablets with internet access (data-enabled SIM cards) were provided to those individuals to enable full participation in the intervention. We have clarified this point in the Methods – Accessibility to the Telehealth Intervention section and revised the Discussion to reflect the importance of supporting digital access in underserved populations. We included this information in the intervention section (page 5): “Participants who lacked access to a suitable device or internet connection were provided with a loaned tablet and internet access (via SIM card) by the research team to ensure equitable participation in the telehealth intervention.”

4. Are the health improvement practitioners also trained in ‘exercise prescription’ or they are promo

---

## [Decision Letter · Decision Letter 1]

7 Oct 2025

Telehealth Exercise Training in Peripheral Arterial Disease (TEXTPAD) Study: A Pilot Randomised Controlled Trial in Socioeconomically Disadvantaged Populations

PONE-D-25-30092R1

Dear Dr. Cucato,

We’re pleased to inform you that your manuscript has been judged scientifically suitable for publication and will be formally accepted for publication once it meets all outstanding technical requirements.

Kind regards,

Enock Madalitso Chisati, PhD

Academic Editor

PLOS ONE

Additional Editor Comments (optional):

Reviewers' comments:

Reviewer's Responses to Questions

**Comments to the Author**

Reviewer #1: (No Response)

2. Is the manuscript technically sound, and do the data support the conclusions?

Reviewer #1: (No Response)

3. Has the statistical analysis been performed appropriately and rigorously?

Reviewer #1: (No Response)

4. Have the authors made all data underlying the findings in their manuscript fully available?

Reviewer #1: (No Response)

5. Is the manuscript presented in an intelligible fashion and written in standard English?

Reviewer #1: (No Response)

Reviewer #1: (No Response)

**Do you want your identity to be public for this peer review?** For information about this choice, including consent withdrawal, please see our Privacy Policy

Reviewer #1: No

---

## [Editor Report · Acceptance letter]

PONE-D-25-30092R1

PLOS ONE

Dear Dr. Cucato,

I'm pleased to inform you that your manuscript has been deemed suitable for publication in PLOS ONE. Congratulations! Your manuscript is now being handed over to our production team.

Kind regards,

on behalf of

Dr. Enock Madalitso Chisati

Academic Editor

PLOS ONE